# A Conceptual Model Proposal to Assess the Effectiveness of IoT in Sustainability Orientation in Manufacturing Industry: An Environmental and Social Focus

Adriane Cavalieri [1,2,*], João Reis [1,3] and Marlene Amorim [1]

1   Department of Economics Management Industrial Engineering and Tourism, GOVCOPP, Campus Universitário de Santiago, University of Aveiro, 3810-193 Aveiro, Portugal; reis.joao@ua.pt (J.R.); mamorim@ua.pt (M.A.)

2   Divisão de Avaliações e Processos Industriais, Instituto Nacional de Tecnologia, Ministério da Ciência, Tecnologia e Inovações, Rio de Janeiro 20081-312, Brazil

3   Industrial Engineering and Management, EIGeS, Campo Grande, Lusofona University, 1749-024 Lisbon, Portugal

*   Correspondence: adriane.cavalieri@int.gov.br

**Abstract:** The scientific literature reveals that there is a gap oriented towards empirical study of the relationship between the Internet of Things (IoT) and sustainability in manufacturing industries. This paper aims to fill this gap by proposing a new conceptual model (CM) for evaluating the effectiveness of IoT technologies in relation to their orientation towards socio-environmental sustainability and the circular economy approach. The research methodology for developing the CM follows the PRISMA protocol, and the data are obtained from the Web of Science (WoS) and Elsevier Scopus databases, focusing on the relationship between IoT and sustainable manufacturing. The PRISMA methodology results in six articles whose statements contribute to the development of the CM. The statements are identified, categorized and organized from the selected articles and divided into dimensions, namely: IoT technology and environmental and social context. The CM incorporates these dimensions and their constructs and indicators to support the assessment of the effectiveness of IoT technologies in relation to socio-environmental sustainability and the circular economy approach. The result of this study is a CM whose objective is to guide organizations in the use of IoT technologies applied to the production and supply chain, in order to create advances in the field of sustainability and the circular economy. The CM will be validated and applied in a manufacturing industry in the next publication. The paper contributes to management practices as it explores the knowledge of performance measurement and evaluation in the context of IoT, sustainability and the circular economy approach.

**Keywords:** Internet of Things; sustainable manufacturing; environmental sustainability; social sustainability; circular economy; conceptual model; performance measurement and assessment system; production process; supply chain

## 1. Introduction

Individuals, society and governments are pushing the manufacturing industry towards the triple bottom line (TBL), which relates to social, economic and environmental sustainability, aiming to protect present and future generations. A sustainable orientation must define the commitment of the manufacturing industries, where the approach of eco-friendly processes, products and services is no longer sufficient from an economic sustainability perspective [1].

Manufacturing can be divided into discrete manufacturing, process and service industries, including activities from customer to factory and vice versa, throughout the manufacturing chain. Manufacturing is of prime importance for the maintenance of the

quality of human life and for service and product delivery, contributing to the world economy. On the other hand, from the point of view of environmental and social sustainability, manufacturing has a major impact on the ecosystem and on working conditions, taking into account the consumption of raw material and energy, the greenhouse effect, the generation of waste, the release of toxic materials, floating plastic and product end-of-life implications [2].

The TBL was proposed by Elkington in an article published in 1998 [3]. The author argues that manufacturing industries should focus on the relationship between economic performance (such as measures of the company's financial performance), environmental performance (aspects such as minimizing environmental waste and improving efficient consumption of resources) and social performance (relating to the well-being of employees and the community) [4]. The approach that develops manufacturing industries towards the TBL is known as sustainable manufacturing (SM) [5].

The definition of SM provided by the US Department of Commerce on the OECD website is: "Manufacturing processes that minimize negative environmental impacts, conserve energy and natural resources, are safe for employees, communities and consumers and are economically sound" [6] (p. 1). However, the Organization for Economic Co-operation and Development (OECD) notes that there is no single common definition of SM. This is an approach oriented towards reducing business risks in any manufacturing operation and maximizing the opportunities that arise from improvements in its processes and products [7]. The OECD website also mentions that in 2011, a Sustainable Manufacturing Toolkit was launched that aimed "to provide a practical starting point for businesses around the world to improve the efficiency of their production processes and products in a way to contribute to sustainable development and green growth" [7] (p. 1).

Since manufacturing is the source of all goods for life, transportation, entertainment, production, safety and health, SM is one of the most important issues for sustainable development [2]. The implementation of SM depends on the alignment of the strategic business plan with the balance between social, economic and environmental sustainability to achieve real benefits from its adoption. However, most of a company's focus is on monetary gains, without a commitment to environmental protection and societal well-being [5].

Despite the fact that economic performance continues to be the dominant objective of companies, they have begun to realize the importance of social issues from the perspective of sustainability. For this reason, they adopt specific practices and measures to obtain a competitive advantage within that perspective, using methods such as customer management, the sharing of information, the practice of transparency and traceability, corporate sustainability reports, corporate social involvement, standardization and monitoring, life cycle assessment, well-being and equity of employees, job stability, sustainable supplier management and the sustainable development of partners [4].

In fact, the technology used in sustainable manufacturing enhances sustainability and represents the most positive contribution to environmental, social and economic prospects [2]. In the context of recent literature, the advent of Industry 4.0, made possible by digital technologies such as the Internet of Things, big data and analytics and artificial intelligence, is perceived as a facilitator of sustainable manufacturing practices and sustainability performance [8–10]. On the other hand, the authors in [8–10] also highlighted some environmental and social sustainability gaps.

There are only a few peer-reviewed articles that have explored Industry 4.0 from a sustainability point of view, highlighting themes such as critical success factors for environmentally sustainable manufacturing, developing a framework for a sustainable Industry 4.0, digitization and the circular economy and cloud manufacturing as an alternative for sustainable manufacturing [10]. Some studies that refer to the relationship between digital technologies (Internet of Things, cloud computing, big data and analytics) and sustainability have inconsistent conclusions, mainly in terms of environmental sustainability, as they have mostly been based on a qualitative view [9].

There is a lack of studies on the effect of each Industry 4.0 technology on the circular economy approach, with respect to themes such as input reduction, consumption, reuse, recovery, recycling and waste and emissions reduction [8]. Moreover, researchers should explore the benefits and challenges within organizations using open-ended or multiple-choice research questions on how to implement and adopt the Internet of Things (IoT) in supply chains, capturing the practical insights of practitioners who are directly involved in IoT adoption and operations [11]. Future work should limit data collection by applying the terms IoT and sustainable intelligent manufacturing, waste valorization, circular supply chain management, zero waste, sustainable manufacturing and/or waste management, targeting a specific industrial sector [12]. There is a lack of knowledge and a limited number of publications dealing with performance measurements in relation to Industry 4.0 [13].

Therefore, the present study aims to fill these gaps by providing a conceptual model (CM) whose objective is to assess the effectiveness of the IoT orientation towards environmental and social sustainability in the operations of manufacturing industries and/or in their supply chains. In this regard, the research question to be discussed concerns how the scientific literature related to IoT, socio-environmental sustainability and the circular economy contributes to the development of the CM with regard to performance measurement and assessment.

The CM is developed via the claims of the scientific literature that discusses the effectiveness of the actions of IoT technologies for environmental and social sustainability. The CM considers the current level of performance in relation to what must be achieved and how it must be achieved, together with what has been assessed and what has not been assessed, in a general way, not measuring performance against a well-defined goal, as highlighted by the survey by Melnyk [14]. The intent of the assessment is to help the organization to engage employees in realizing the performance of IoT technologies. The evaluation focuses on environmental and social sustainability and should not include many restrictions on how to achieve these goals, given the dynamics and the turbulent environment that may lead to the determination of new measures [14]. Therefore, the assessment is oriented towards the progress and success of sustainability within the business context.

This paper contributes to knowledge about the measurement and evaluation of performance in relation to the effect of IoT technologies on sustainability and circular economy approaches in production operations and supply chains. The CM should help the organization to engage employees in the assessment and measurement of the effectiveness of IoT technologies, with a focus on socio-environmental sustainability and the circular economy.

The paper is organized as follows. Section 2 describes the research methodology for developing the CM related to "IoT and sustainable manufacturing". Section 3 presents the results as a mapping of the selected sources by PRISMA (preferred reporting items for systematic review and meta-analysis), a categorization of the statements of the selected sources, a synthesis of the statements and the CM itself. Section 4 presents a discussion of the results. Section 5 presents the theory and managerial contributions, Section 6 reveals the research limitations and Section 7 presents some suggestions for future research.

## 2. Methodology

The authors of this study acknowledge that the meaning of "conceptual model" comes from the definition of a "conceptual framework" (CF), which is related to a "network" of concepts that are interconnected and together provide a comprehensive understanding of a real-life phenomenon. The concepts that are part of the conceptual framework support one other and articulate their respective phenomena [15]. Jabareen [15] presents the main features of the conceptual framework. The CF is called a conceptual model (CM) when it contains the methodological assumption of assessing the "real world" and uses variables or factors [15].

The study presents a CM that contains the methodological assumption of assessing the "real world" with regard to constructs, factors, indicators and their definitions, focusing

on the effectiveness of IoT technologies for sustainable manufacturing (environmental and social sustainability).

The development of the CM follows a methodology adapted from Jabareen as follows [15]:

1.  Mapping of the selected sources: carrying out a systematic literature review applying the PRISMA process, focusing on the IoT and sustainable manufacturing (environmental and social sustainability) and the identification of contents or statements related to empirical facts and practices.
2.  Categorizing the selected sources:

    2.1  Extensive reading and categorization of selected content by reading selected literature and categorizing the content into dimensions and representative constructs with each dimension.

    2.1.1  Identify and name concepts: review selected content allowing concepts to emerge from the literature.

    2.1.2  Deconstruct and categorize the concepts: identify the main attributes, characteristics, assumptions and roles of concepts and, later, organize and categorize them according to their features.

    2.2  Integration of concepts: integrate and group concepts that have similarities to a new one, manipulating the concepts to give a reasonable number.

3.  Synthesis, resynthesis and making sense: conducting an iterative process that includes repetitive synthesis and resynthesis, until the researcher recognizes a CM that makes sense.

The methodology is illustrated in Figure 1.

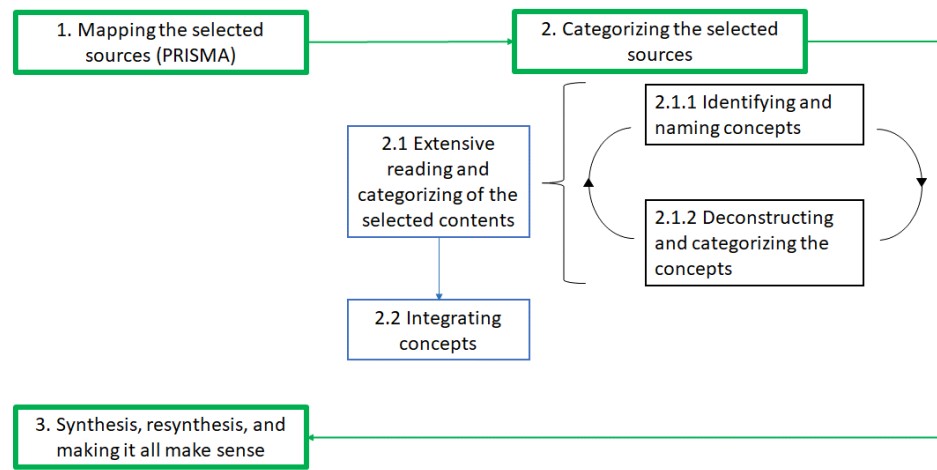

**Figure 1.** CM development methodology.

## 3. Results

### 3.1. Mapping the Selected Sources: PRISMA

The publications eligible for the design of the CM were mapped via a systematic literature review using the PRISMA process (preferred reporting items for systematic reviews and meta-analyses) [16]. PRISMA employs structured and explicit methods, with the following phases: (1) identification phase, (2) screening phase, (3) eligibility phase and (4) inclusion phase, to identify, select and critically evaluate the relevant research. The main benefit is to minimize bias, as this can make it difficult to conduct and interpret the review.

The article by Malek and Desai [5] presents a systematic literature review to map the literature in relation to sustainable manufacturing. These authors adopted the terms ("Sustainable Manufacturing" OR "Sustainable Production" OR "Sustainable Operations") as keywords to select the publications. Following their work, this research considered the

same keywords for the Web of Science (WoS) and Elsevier's Scopus databases to ensure the identification of high-quality scientific articles on sustainable manufacturing. The research on WoS and Scopus was conducted on 14 April 2022.

### 3.1.1. Identification Phase

The articles were identified through the WoS and Scopus databases. The search began in WoS and Scopus with the exact phrases in "TOPIC" or "TITLE-ABS-KEY" ("Sustainable Manufacturing" OR "Sustainable Production" OR "Sustainable Operations") AND ("Internet of Things"), resulting in 82 (eighty-two) documents and 416 (four hundred and sixteen) documents, respectively.

### 3.1.2. Screening Phase

An additional refinement was made in relation to the types of documents, to maintain the quality of the present study, including journal articles, early access and reviews without defining the range of the years of the publications and considering only articles in English. The application of these criteria resulted in 51 (fifty-one) articles (WoS) and 275 (two hundred and seventy-five) articles (Scopus).

Further refinement was performed by considering WoS's subject categories. Thus, this stage generated 26 (twenty-six) articles, since the following categories were excluded: Agronomy; Chemistry Analytical; Chemistry Multidisciplinary; Electrochemistry; Energy Fuels; Engineering Chemical; Engineering Electrical Electronic; Engineering Mechanical; Food Science Technology; Instruments Instrumentation; Materials Science Multidisciplinary; Mathematics; Mathematics Interdisciplinary Applications; Physics Applied; Telecommunications. The selected categories and respective numbers of articles were Automation Control Systems (1), Computer Science Interdisciplinary Applications (4), Engineering Environmental (4), Engineering Industrial (6), Engineering Manufacturing (9), Environmental Sciences (10), Environmental Studies (5), Green Sustainable Science Technology (9), Management (2), Multidisciplinary Sciences (1) and Operations Research Management Science (7).

Further refinement was performed by considering the Scopus subject areas. Thus, this stage generated 80 (eighty) articles, as the following categories were excluded: Agricultural and Biological Sciences; Arts and Humanities; Biochemistry, Genetics and Molecular Biology; Chemical Engineering; Chemistry; Decision Sciences; Earth and Planetary Sciences; Economics, Econometrics and Finance; Energy; Health Professions; Materials Science; Mathematics; Medicine; Multidisciplinary; Physics and Astronomy; Psychology; Social Sciences. The selected categories and respective numbers of articles were Business, Management and Accounting (7), Computer Science (54), Engineering (55) and Environmental Science (6).

### 3.1.3. Eligibility Phase

The abstracts were checked. The last refinement criterion was applied in relation to the abstracts of the articles, where the selection was based on articles related to the "production process" and "environmental–social" sustainability.

From WoS, this step generated 5 (five) articles that were read in their entirety. However:

- Two articles were discarded: one was related to life cycle assessment (LCA) and other article concerned lean manufacturing and healthcare. Both subjects were out of the scope of this work.

From SCOPUS, this step generated 11 (eleven) articles that were read in their entirety. However:

- Three articles were duplicates of articles in WoS. Two of these were discarded as mentioned earlier, and the article by Li et al. [17] was selected to contribute to the study.
- Five articles were not considered in the research. Two articles were discarded, as one was concerned with manufacturing process modeling and other was out of the industry context. One article was not free and another proposed the design of a

business maturity scheme for companies implementing Industry 4.0. Finally, another article focused on manufacturing resiliency and sustainability.

### 3.1.4. Inclusion Phase

Suitable full texts should be selected to contribute to the qualitative synthesis of the research. A total of 6 (six) articles were selected to contribute to the design of the CM for manufacturing organizations.

The PRISMA results are shown in Figure 2.

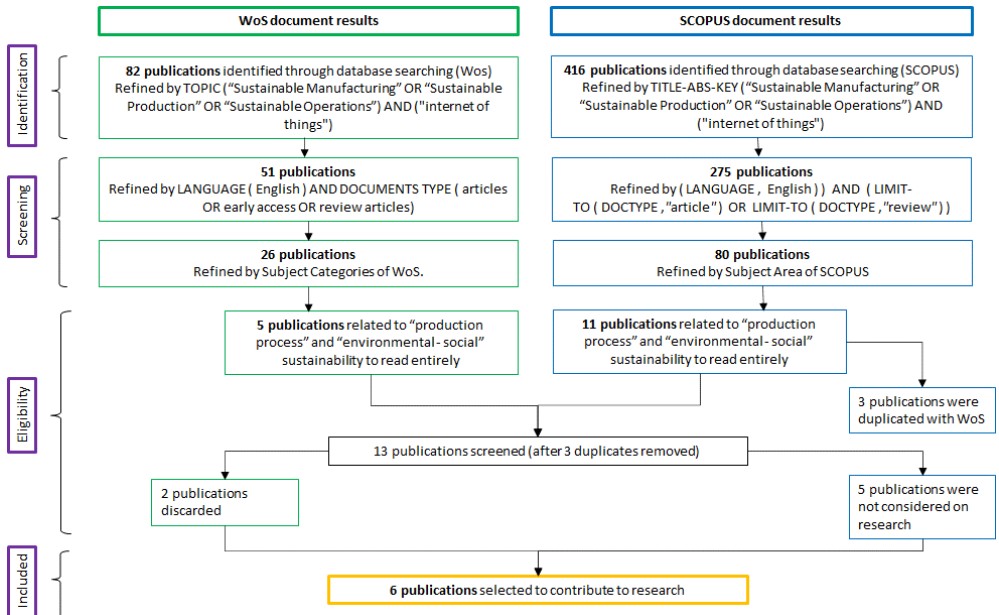

**Figure 2.** PRISMA process: (1) identification phase; (2) screening phase; (3) eligibility phase; (4) inclusion phase.

The selected sources supported the lack of a comprehensive understanding of Industry 4.0 technologies and sustainable manufacturing in relation to an empirical point of view, as demonstrated by their suggestions for future research, for example, the development of a model to assess simultaneously the capabilities of both sustainability aspects and Industry 4.0 [18], the relationship between organizational performance and digitalization of environmental sustainability practices [19], exploring the impact of enabling technologies on sustainability pillars for manufacturing industries in developing countries [20] and research on the effects of the emerging technological field of Industry 4.0 on sustainable manufacturing [21].

### 3.2. Categorizing the Selected Sources

The phase of categorizing the selected sources was developed with the support of Cavalieri's research [22,23].

The PRISMA methodology resulted in six articles, in which the statements were related to empirical facts and practices regarding IoT, social–environmental sustainability and the circular economy approach. The statements were identified, categorized and organized according to their attributes into dimensions and their representative constructs, and then grouped into factors to be assessed by the indicators, as follows:

- Dimensions: the three principal dimensions to be assessed were IoT technology, environmental approaches and social approaches.
- Constructs:

i.  The seven constructs for IoT technology were: IoT expectations, IoT technology capacity, IoT technologies integration, IoT-based process management, IoT data, IoT challenges and IoT barriers.
ii.  The four constructs for environmental approaches were: company engagement, performance measurement methodology, performance indicators implementation and environmental sustainability and circular economy practices.
iii.  The four constructs for social approaches were: company engagement, performance measurement methodology, performance indicators implementation and social sustainability practices.

- Factors to be assessed/measured by the indicators, which should fit the statements and their definitions related to the construct.

### 3.3. Synthesis and Resynthesis: The Statements and Their Dimensions, Constructs and Factors

The integration and grouping process of the statements resulted in dimensions and constructs being categorized and organized according to their attributes. The statements of the selected articles were organized and grouped as follows:

### 3.3.1. IoT Technology

- IoT Technology Expectation

Top management chooses to implement IoT technology because it is expected to be beneficial in the digitalized world or because IoT technology is seen as an agent of improvement or an agent of transformation that perpetually manifests shock waves throughout the organization [19].

- IoT Technology Capacity

The IoT technology supports the management of a collection of large amounts of data for the production process and for data analysis and data mining [17], as a sophisticated and advanced technological device for massive data storage, retrieval, processing and analysis [18]. After transferring large amounts of data through the cloud, data analysis is completed and useful information is produced [20] which can be used to guide production decision-making [17].

The use of IoT technology enables the development of smart manufacturing via the interrelationship between the smart constituent elements of the company, such as smart products, smart facilities (sensors, data storage equipment, software), physical entities (parts, machinery) and networking components (interfaces, wired and wireless network protocols) [17]. The complex manufacturing network supports real-time management through the integration of internal business departments (vertical integration) and/or the integration between the company and the supply chain (horizontal integration) [17].

- IoT Technologies Integration

The "intelligent factory" uses information and communication technologies such as platforms and applications to integrate the production area with other company departments, distributors and customers, giving rise to more transparent supply chains [21]. The digitization and interconnection of industrial processes enabled by IoT technologies is facilitated by data analytics, machine learning and artificial intelligence [21]. Machines can trade and interact to reconfigure themselves for the dynamic nature of production, and the new smart system is integrated with existing systems in a compatible manner [18]. The entire factory can be wirelessly interconnected, monitored and controlled [20].

Production processes are supported by networked IoT technologies for data collection, exchange and analysis [18]. Smart network connections employing IoT technologies are established between different subsystems within the company, including sensors, actuator control, management and the manufacturing area (vertical integration), and different companies in the supply chain can share and exchange information (horizontal integration). Furthermore, different product-oriented (or package) processes can be established through-

out the product life cycle starting from customer needs and product design to product maintenance and recycling (end-to-end integration) [18].

- IoT-Based Process Management

IoT technologies increase cooperation by sharing information and collaboration within the supply chain. They support the monitoring and control of the production process through data collection, providing a reference for making business decisions through big data analysis, allowing information to be obtained and shared to facilitate collaboration between people and things [17].

IoT technologies are also used to understand the status of the supply chain by obtaining real-time information from all nodes in the supply chain. Furthermore, IoT technologies facilitate the interconnection of physical and virtual space (widespread deployment of distributed devices embedded with computing, identification, communication and sensing capabilities) [17].

- IoT Data

IoT technology supports decision-making with high-quality data from customer relationship management, warehousing management, production management and the supply chain, considering the set of interconnected processes that cover the entire product life cycle and the entire company operation [17].

- IoT Challenges

Companies may face difficulties regarding the implementation of the identification and sensing of "smart objects". Sensing technologies include radio-frequency identification (RFID), wireless sensor networks (WSNs), near-field communication (NFC) and Bluetooth technology (BT). In addition, there are difficulties in collecting data for the identification of "smart objects", which are embedded in computing and communication resources, and problems in the management of "smart objects" because they are numerous, heterogeneous and dynamic [17].

Data transfer over network technology is considered a problem for company decision makers. The full operation of the transfer of data by the network technology requires addressing, routing, end-to-end transmission, gateways and traffic characterization. Network technologies depend on wired technologies, wireless technologies, cellular networks and satellite communication technology [17].

- IoT Barriers

There are organizational, technological, governmental and ecosystem maturity barriers [20]. Organizational barriers include lack of a senior management support system, resistance from the top management system, low perceptions regarding the digital revolution, risky investment in technologies, unavailability of a digital strategy, unavailability of a data-based service system and fluctuations in production size. Technological barriers include the high cost of technology, the unavailability of I4.0 standards, the unavailability of a data security system, low IT levels, the unavailability of IT infrastructure and service centers and the quantity of parts to be produced. The governmental and ecosystem maturity barriers include the unavailability of government policies, lack of support from government, the low maturity of the manufacturing industry, the unavailability of a technology ecosystem and the lack of consultants and trainers in the area.

### 3.3.2. Environmental Approach

- Company Engagement

Environmental challenges and the concern for sustainability are key issues that companies consider when developing their strategy [21]. A reliable definition for sustainable manufacturing practice develops sustainability awareness among companies and their supply chains, and between companies and their customers [18]. Environmental sustainability

principles should be incorporated into companies' business models to help them understand whether environmental sustainability initiatives could lead to better performance, regardless of social responsibility aspects [19].

An integrated smart and sustainable business model can be based on company-specific strategies [18] with regard to the relationship between environmental sustainability and digital transformation [19]. Digital technologies offer organizations opportunities to develop new business models focused on the environment, incorporating environmentally sustainable practices justified by digital transformation [19], for example, the company may rely on the IoT for carrying out sustainable business practices in relation to reducing carbon emissions and minimizing solid waste discarded into the environment [19].

Digital transformation is a hot topic for discussion among top-level management, with respect to how environmental sustainability practices can become a part of the strategic decision-making process [19]. Consumers demand "environmentally friendly" products, and companies perceive new business possibilities through Industry 4.0 technologies [21].

- Performance Measurement Methodology

Digital technologies are used to develop new performance measurement methods for "sustainable and smart manufacturing" [18]. The sustainability performance of IoT manufacturing should follow a guideline and a standard for environmental sustainability assessment metrics, which should be agreed among employees [18].

The existing tools and methods establish a reliable approach for "environmental sustainability and smart manufacturing" [18]. Data-driven smart algorithms focus on sustainable manufacturing, sustainable supply chains and sustainable product end-of-life and life cycle assessments [18]. Different types of sensors result in the development of specific performance criteria to mitigate negative effects on the environment without detriment to competitiveness [21].

- Performance Indicators Implementation

Clean manufacturing processes, driven by digital technology, can reduce costs without harming the environment and without negative impacts on the ecosystem [19].

Smart business processes, which rely on cleaner and more sustainable mechanisms seen from the economic, environmental and social points of view, may offer several favorable circumstances simultaneously, such as can reducing operating costs, improving profitability and shop-floor employee safety and reducing the environmental impact of the business [19]. In addition, seen from the economic and environmental point of view, they may offer increased production rates, effective utilization of resources, reduction of carbon dioxide ($CO_2$) emissions and waste reduction [20].

The IoT represents an opportunity to drive sustainable manufacturing, enabling the use of environmentally friendly, abundant and locally available resources [18]. The IoT enables all types of data collection and analysis from industrial processes, easily helping to avoid unnecessary manufacturing steps [21]. IoT technologies allow the company to improve the impact of the process on the environment, eliminating waste throughout the value chain, enhancing sustainable consumption, eliminating harmful waste discarded into the environment [19], reducing scrap on the shop floor [20] and contributing to reducing the entry of virgin resources, the generation of waste [18,21] and $CO_2$ emissions [20].

The interconnection of processes allowed by IoT technologies causes an increase in the development of performance indicators [21]. The company obtain higher-quality data generated from IoT to support decision makers with information from production management on topics such as raw materials, energy consumption, water consumption, water waste, solid waste, by-products [17], the use of packaging [21] and air pollution [19], and the reduced consumption of materials results in less dependency on natural resources [20].

Power consumption can be reduced due to the improved precision of data monitoring via IoT technologies [20,21]. On the other hand, data centers consume large amounts of energy and resources, impacting the environment negatively, as monitored by IoT technologies [24]. There is an additional environmental liability measurement associated

with Industry 4.0 as a consequence of the materials required for electrical devices, which are sometimes scarce and may require intensive extraction and processing efforts, which may negate the environmental advantage of the Industry 4.0 context [18].

- Environmental Sustainability and Circular Economy Practices

Digital technologies are used to develop new ways of coping with waste [19]. IoT technologies improve manufacturing process efficiency regarding the 6R design ("reduce", "reuse", "recycle", "recover", "redesign" and "remanufacture"), in order to save natural resources [18,20]. Regarding the improvement of the economic and ecological flows of resources, IoT technologies enable collaboration and partnerships among a company's stakeholders with respect to "closing the loops" by reusing raw materials, sharing raw materials, reusing waste, sharing waste [21] and allowing the reutilization of materials in a remanufacturing process [20]. Industry 4.0 technologies assist in raw materials purchasing from suppliers when needed (the raw material or semi-finished production material is requested on demand) [24].

The incorporation of different types of sensors allows greater transparency of operations, which adds intelligence to processes to mitigate negative effects on the environment and throughout the supply chain, reducing the losses generated along the entire chain [21]. The IoT supports information and communication technologies such as platforms and applications that are employed within the "intelligent shop floor", helping to reduce energy consumption, solid waste, the use of packaging, by-products [21], the use of raw materials, water consumption and water waste [17]. These technologies are employed to integrate the production area with distributors and customers, helping to reduce waste, energy and the use of packaging [21]. The IoT sensors provide real-time monitoring information for better air quality [19].

The utilization of IoT technologies ensures a dynamic interconnection among energy providers, the company and market demand, which leads to better energy management [18]. Continuous monitoring through smart devices increases the visibility and awareness of energy consumption by using real-time problem solving [21].

3.3.3. Social Approach

- Company Engagement

System integration promotes communication between different levels of the company (and between manufacturing plants), which supports the development and strength of the company's values and corporate culture [21].

A digital culture, with associated training, may be a challenge for companies [20], especially at the early stage of Industry 4.0 implementation. Many people are afraid that digital solutions and digital technologies may result in a loss of jobs [20]. Some employees have lost their jobs due to insufficient knowledge of digital technologies [20], and some employees have damaged sensors and interface devices or refused to follow the instructions [20]. On the other hand, there are digital technology solutions developed by companies to address career sustainability issues arising from machines replacing humans [18] and to ensure a safer workplace, leading to a decrease in workplace accidents and an increase in employee morale, as well as making work easier [20].

The increased use of digital technologies creates new jobs with a different profile [20], where the major challenge is finding and retaining creative people and people with strong analytical skills [20]. As digital technologies develop dynamically, training keeps employees up to date [20].

- Performance Measurement Methodology

There should be a guideline for determining the performance of smart industries with regard to social sustainability [18]. The existing tools and methods should establish a reliable approach for "social sustainability and smart manufacturing", together with standard social sustainability assessment metrics regarding the IoT technologies implementation

agreed upon by the company, as mentioned by Sartal [21] in relation to environmental sustainability.

- Performance Indicators Implementation

Employees should continuously adapt to the new restrictions imposed on job options due to industrial innovation, and they should be able to understand the information and use information from various sources related to various subjects to maintain a career and upgrade their abilities to perform new tasks, via continuous education [18].

New skilled and trained employees are needed to apply IoT technologies effectively [18]. New profiles of employees are immediately required for positions related to the application of digital technologies [21]. Manual work is reduced in favor of cognitive and analytical skills, fundamentally linked to information technologies and data analysis [21].

- Social Sustainability Practices

IoT technologies trigger the development of jobs in different areas, for example, automation engineering, control system configuration, artificial intelligence and software engineering, as well as reducing most types of lower-skilled jobs [20].

Smart grids allow machines to communicate and make small decisions without human intervention [18]. Machine communication and negotiation pave the way and increase the demand for new jobs, where humans are focused on designing, developing and maintaining this network of machines [18].

IoT technologies offer employees better and safer working conditions [18]. The IoT helps to improve equipment and operator safety through better maintenance solutions and by providing real-time hazard warnings [21].

In the smart shop floor, machines work hand in hand with humans, observing them and learning from them in a way very similar to an apprentice, complementing humans rather than replacing them, offering labor career sustainability [18]. However, it is still argued that IoT technologies lead to a shrinking of the human workforce, thus reducing job opportunities and increasing unemployment. This may result in resistance against adopting Industry 4.0 initiatives [18].

A company can implement big-data-driven systems and IoT technologies for a better division of labor between humans and smart machines, to address the security, privacy and ethical issues introduced by smart manufacturing networks [18]. Digital technologies allow collaborative networks for employees (shop-floor employees and managers) to exchange their knowledge and experiences with the supply chain [21], as typical activities in any manufacturing environment (e.g., analysis, cooperation, creativity) continue to be carried out by human workers [21].

The synthesis of the statements and their dimensions, constructs and related factors to be assessed/measured are summarized in Figure 3.

### 3.3.4. The Conceptual Model

The CM provides an interpretative approach to the current state of the art described in the literature, where the systematic literature review plays a key role in establishing knowledge. The iterative and repetitive synthesis process resulted in the following dimensions, constructs, indicators and indicator definitions (see Tables 1–3).

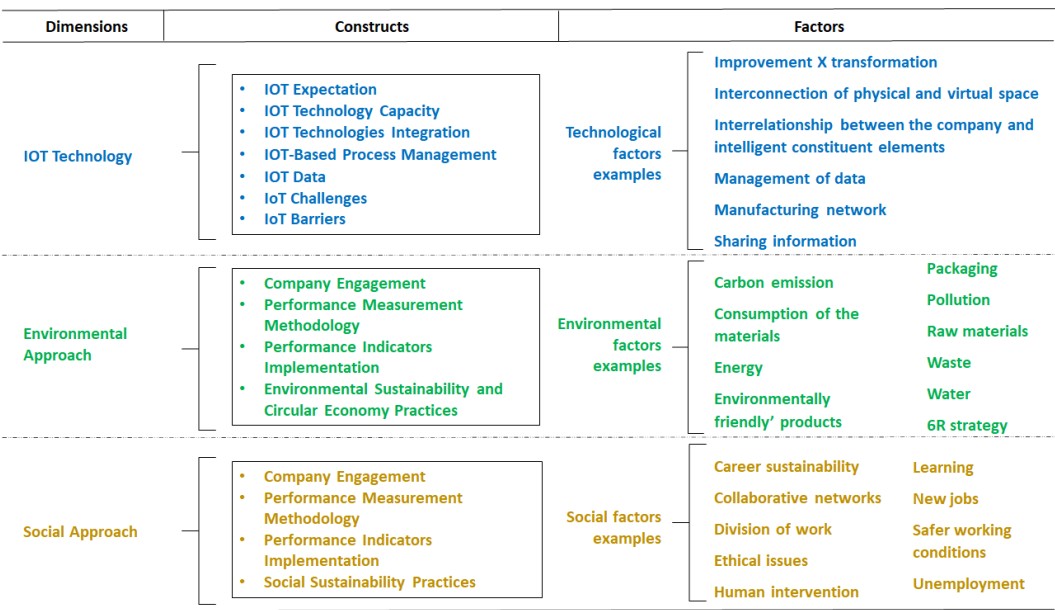

**Figure 3.** The synthesis of the statements and their dimensions, constructs and factors.

**Table 1.** IoT Technology.

| Dimension | Construct | Indicators | Indicator Definitions | Source |
|---|---|---|---|---|
| IoT Technology | IoT expectations | • Company leadership's view of IoT. | • The viewpoint of top managers and managers in relation to IoT technology perspectives, which influences their purpose of IoT implementation. | [19] |
| | IoT technology capacity | • Level of technological sophistication.<br>• Level of manufacturing network.<br>• Level of data management.<br>• Interrelationship among the company's intelligent constituent elements. | • The company's potential to innovate regarding IoT technologies within its operations and in its supply chain. | [17,18,20] |
| | IoT technologies integration | • Connections among the company's intelligent network.<br>• Networked processes access and use.<br>• Compatibility of the digital system.<br>• Level of shop-floor interconnection. | • The company's intelligence regarding the interconnection of the IoT technologies within its operations and in its supply chain, which enables the identification, selection, analysis and management of relevant information on potential events or problems with real-time responses. | [18,20,21] |
| | IoT-based process management | • Level of IoT contribution to process management (production–company supply chain) | • The company's purpose of IoT implementation within its operations and in its supply chain. | [17] |

**Table 1.** *Cont*.

| Dimension | Construct | Indicators | Indicator Definitions | Source |
|---|---|---|---|---|
| | IoT data | • Level of data route between the internal and external company processes considering customer, warehousing, brewhouse process, fermentation process, beer processing and supply chain. | • The company's data flows regarding IoT technology within its operations and in its supply chain to support decision makers, which influences the performance measurement implementation. | [17] |
| | IoT challenges | • Data access and utilization of "smart objects". <br> • Transference and data analysis results. | • The company's issues when operationalizing the IoT. | [17] |
| | IoT barriers | • Organizational/technological/ governmental and ecosystem maturity | • The difficulties that company faces in implementing IoT. | [20] |

**Table 2.** Environmental Approach.

| Dimension | Construct | Indicators | Indicator Definitions | Source |
|---|---|---|---|---|
| Environmental Approach | Company engagement | • Development of environmental sustainability strategy. <br> • Development of environmental sustainability and digital transformation integration strategy. <br> • Development of environmental sustainability and digital transformation politics. <br> • Definition of sustainable manufacturing practices. <br> • Development of new business model via environmental sustainability and IoT integration. <br> • Implementation of sustainable manufacturing and IoT technologies integration practices. | • The company commitment to environmental sustainability and digital transformation relationships in its own operation and in its supply chain. | [18,19,21] |
| | Performance measurement methodology | • Development of environmental sustainability assessment methodology. <br> • Guidelines for the environmental sustainability performance. <br> • Definition of tools and methods for "environmental sustainability and smart manufacturing" performance. <br> • Development of data-driven smart algorithms for design of "environmental sustainability and smart manufacturing". | • The development of procedures to measure the resource consumption (water use, raw materials, energy), the design for 6R, by-products and the pollution generated by the company and its supply chain based on IoT technologies. | [18,21] |
| | Performance indicators implementation | • Environmental sustainability assessment metrics. <br> • Performance measurements for "environmental sustainability and smart manufacturing". <br> • Performance metrics for additional environmental liability. | • The assessment of the resource consumption (water use, raw materials, energy), the design for 6R, by-products and the pollution generated by the company and its supply chain based on IoT technologies. | [17–21,24] |

**Table 2.** *Cont.*

| Dimension | Construct | Indicators | Indicator Definitions | Source |
|---|---|---|---|---|
| | Environmental sustainability and circular economy practices | • Platforms and applications employment within the company and in its supply chain.<br>• Environmental sustainability and smart manufacturing design for 6R.<br>• Records of environmental sustainability and smart manufacturing improvement process.<br>• Monitoring information of the operations and the supply chain through smart devices. | • The digital technology (IoT technologies) and environmental approach as integrated practices to mitigate and/or eliminate resource consumption (water use, raw materials, energy), by-products and pollution, and to improve the design for 6R, by the company and its supply chain. | [17–21,24] |

**Table 3.** Social Approach.

| Dimension | Construct | Indicators | Indicator Definitions | Source |
|---|---|---|---|---|
| Social Approach | Company engagement | • Development of social sustainability strategy.<br>• Development of training program for digital technologies.<br>• Development of program for career sustainability.<br>• Politics for a safer workplace.<br>• Politics for job retention.<br>• Politics for bilateral communication. | • The company commitment to social sustainability and digital transformation relationships in its own operation and in its supply chain. | [18,20,21] |
| | Performance measurement methodology | • Development of social sustainability assessment methodology.<br>• Definition of tools and methods for the "social sustainability and smart manufacturing" performance.<br>• Guidelines for the social sustainability performance. | • The development of procedures to measure human factors in relation to IoT technologies implementation in the company and its supply chain. | [18,21] [1] |
| | Performance indicators implementation | • Valorization of worker adaptation.<br>• Valorization of self-learning.<br>• Valorization of continuous learning.<br>• Assessment metrics for job sustainability.<br>• Assessment of the new skills and new profiles.<br>• Implementation of social sustainability and smart manufacturing performance metrics. | • The assessment of human factors in relation to IoT technologies implementation in the company and its supply chain. | [18,21] |
| | Social sustainability practices | • Valorization of cognitive and analytical human skills.<br>• Valorization of the human work.<br>• Human autonomy and mediation.<br>• New kind of working conditions.<br>• Concerns regarding unemployment.<br>• Collaborative networks.<br>• Bilateral communication. | • The digital technology (IoT technologies) and social approaches as integrated practices to boost the human factors in the company and in its supply chain. | [18,20,21] |

[1] This indicator definition is an insight from reference [21].

## 4. Discussion

An effective way of monitoring and evaluating performance is the introduction of key performance indicators (KPIs) that fit the strategic intentions of the company [25]. Performance measures applied with Industry 4.0 technologies should be able to capture local contexts and a wide range of phenomena from the external context and analyze a large amount and variety of data [26]. In this regard, performance measures should be

autonomous and heterogeneous in detecting data and planning information to support the management of the production process and the supply chain [26].

IoT technology is an example of an Industry 4.0 technology that is employed to achieve smarter manufacturing and performance measurement. It is believed to be a critical step in industry [27]. The use of IoT for shop-floor management is facilitated by the fact that the technology can be installed in a limited area such as on a production line, in a storage area or on a packing line [27].

IoT technologies are generally employed to monitor environmental conditions in industry, for example using low-cost sensors to collect the necessary data within production facilities, e.g., "temperature, humidity, air pressure, air quality (carbon monoxide, liquid petroleum gas, smoke), lightning and noise" [25] (p. 286). Some requirements should be met to make these solutions viable, consisting of the three constraints of scalability, adaptability and cost-effectiveness. Scalability refers to the applicability to different sizes of installations, as well as the subsequent adjustment to any size changes. Adaptability means that the system should be easily adaptable according to the prevailing environmental conditions in the production area, through a quick and simple exchange of the employed sensors. Another requirement is cost-effectiveness, for example using common sensors in the plant [25].

Most companies still aim for monetary gains without a commitment to environmental protection and the well-being of society. The main purpose of KPIs in production systems is to improve quality and efficiency, reduce costs and lead times and increase flexibility and profitability. Results are presented quantitatively and compared with the performance target to understand how plant productivity should be increased [13].

Some researchers are applying the ISA-95 and ISO-22400 standards as a support for developing and implementing a performance measurement system referring to the IoT-based production performance model [13,27], in relation to the definition of manufacturing processes and performance indicator formulas, respectively.

ISO-22400 defines the application of KPIs and sub-KPIs and their formulas, corresponding elements and benefits, from thirty-five KPIs oriented towards a manufacturing execution system (MES)—a software package combining multiple execution management components into single and integrated solutions focused on the management of shop-floor operations such as material delivery and consumption, as well as production progress [27]. On the other hand, the ISA-95 standard describes entities at the shop-floor level, where information technologies and operation technologies interact, developing an automated interface between the company and the control systems [13]. In this regard, the IoT device detects a product on the production line, and its data such as "time", "quantity", "location", "value" and "status" are sent to the MES. Then, these data are aligned with the production performance model, which consists of the three subparts of actual equipment, actual produced material and actual consumed material [27].

The CM contributes to emphasizing the balance between social and environmental sustainability in the business context to achieve the benefits of adopting sustainable manufacturing in the production process and supply chain; that is, it goes beyond monitoring the environmental conditions of the industry, as mentioned earlier. In other words, the CM is concerned with reducing or eliminating environmental hazards, to preserve natural resources and employee/community well-being.

The objective of the CM is to assess the performance of the strategy at the shop-floor level and in the corresponding supply chain, considering the IoT technology dimension, the environmental sustainability dimension and the social sustainability dimension.

For instance, in relation to the IoT technology dimension, the CM aims to assess the point of view of managers and top managers in relation to IoT technology perspectives, which influences their IoT implementation purposes. The company's potential for innovation in IoT technologies lies in its operations and supply chain. The company's "smartness" and "responsiveness" also lies in the interconnection between the IoT technologies of its operations and supply chain, which allows the identification, selection, analysis and

management of relevant information regarding potential events or problems with real-time responses. Thus, the company's data flows regarding IoT technology, within its operations and across its supply chain, support decision makers and influence the performance measurement implementation.

With regard to the dimension of environmental sustainability, the CM proposes to assess the company's commitment to the relationship between environmental sustainability and digital transformation in its own operation and in its supply chain. At this level, the development of procedures should be carried out to measure resources consumption (i.e., usage of water, raw materials and energy), the design for 6R, by-products and the pollution generated by the company and its supply chain, based on IoT technologies. Digital technology (IoT technologies) and environmental approaches as integrated practices must exist to mitigate and/or eliminate resource consumption (water use, raw materials and energy), by-products and pollution, and to improve the design for 6R, by the company and its supply chain.

Regarding the dimension of social sustainability, the CM recommends assessing the company's commitment to social sustainability and digital transformation in its operations and in the supply chain. It is also related to the development of procedures to measure human factors in relation to the implementation of IoT technologies in the company and its supply chain, as well as the assessment of the human factors in relation to the implementation of IoT technologies in the company and its supply chain. Digital technology (IoT technologies) and social approaches as integrated practices enhance human factors in the company and its supply chain.

Finally, the IoT technology dimension is directly related to the environmental and social sustainability dimensions that affect the results of the associated indicators. In addition, these indicators must be aligned with the KPIs oriented towards the productivity category in the production process and in the supply chain. For example, we refer to the ISA-95 and ISO-22400 standards, economic sustainability, the transformation to the triple bottom line (TBL) and circular economy approaches.

## 5. Theoretical and Managerial Contributions

This paper contributes to the scientific literature regarding:

- The measurement and evaluation of performance in Industry 4.0 within the scope of sustainability and the circular economy.
- The effect of IoT technologies in approaching the circular economy on themes such as input reduction, consumption, reuse, recovery, recycling and waste and emissions reduction.
- Exploring the benefits and challenges within organizations on how to implement and adopt the IoT and environmental and social sustainability in its operations and supply chains.

This paper contributes to the business and management context regarding:

- The CM, which should help the organization to engage employees in assessing the effectiveness of IoT technologies with a focus on socio-environmental sustainability.
- The focus on socio-environmental sustainability, which can lead to new or revised measures that improve the organization's sustainability performance.

## 6. Research Limitations

The limitation of this research is that the CM was not validated, used as a management tool to assess the real world, or applied as a pilot study.

## 7. Suggestions for Future Research

The suggestions for future research are twofold, as they allow mitigation of some of the limitations of this article but also provide new and exciting avenues of research, as follows:

- The validation of the CM, according to Jabareen [15] (p. 54), to certify "whether the proposed framework and its concepts make sense not only to the researcher but also to other scholars and practitioners ( . . . ) is a process that starts with the researcher, who then seeks validation among 'outsiders'". The researches can receive feedback, new insights and comments from expert opinions—the "outsiders"—by consensus methods such as the Delphi Method "to gather general agreement on topics that do not yet have empirical evidence to support future decisions or actions; often, these topics are ambiguous or controversial" [28] (p. 663). The Delphi method has advantages for obtaining consensus from other methods as "it eliminates the bias and influence that can occur in face-to-face meetings as the respondents are to remain anonymous, ( . . . ) the ranking of each item by the entire response group helps make the ultimate conclusions more reliable than a single meeting, ( . . . ) does not require specified meeting times" [28] (p. 666).
- The translation of the statements into a management tool to assess whether IoT technologies are oriented towards socio-environmental sustainability and circular economy approaches in manufacturing industries. The management tool can be a questionnaire using a five-point Likert scale via the online platform SurveyMonkey and/or interviews. The results should contribute to the management practices as an input to the planning and implementation of IoT technologies oriented towards sustainability and the circular economy approach. In addition, the results contribute to the scientific–academic environment, because there is a demand for empirical studies on, for example, the impacts of digital transformation on the environmental and social domains of sustainability, the relationship between organizational performance and digitalization of environmental sustainability practices and digital transformation strategies integrated with sustainability pillars in manufacturing.
- The application of this management tool in a pilot study aimed at a specific industrial sector to investigate, for example, the differences in the socio-environmental effectiveness of the IoT in small, medium and large organizations in a certain sector.

**Author Contributions:** Conceptualization, A.C.; methodology, A.C.; formal analysis, A.C.; investigation, A.C.; writing—original draft preparation and editing, A.C.; review, J.R. and M.A.; supervision, J.R. and M.A. All authors have read and agreed to the published version of the manuscript.

**Funding:** This research received no external funding.

**Institutional Review Board Statement:** Not applicable.

**Informed Consent Statement:** Not applicable.

**Data Availability Statement:** Data sharing is not applicable to this article.

**Acknowledgments:** Adriane Cavalieri would like to register that she appreciates the support of the University of Aveiro in enabling postdoctoral work, on behalf of João Reis and Marlene Amorim.

**Conflicts of Interest:** The authors declare no conflict of interest.

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
