# Peer review of "A Conceptual Model Proposal to Assess the Effectiveness of IoT in Sustainability Orientation in Manufacturing Industry: An Environmental and Social Focus"

_applsci, doi:10.3390/app12115661_

Round 1

Reviewer 1 Report

  1. The contribution of the paper is missing. Please clearly itemize the paper contribution in the introduction section.
  2. The figures' titles are ambiguous and need more specified explanations.
  3. For the proposed conceptual model to be more effective, some experimental experiences are required, such as information gathered from questionnaires completed by stakeholders.

Author Response

General comments

First, we would like to thank you for the time devoted in providing insightful recommendations. We believe that your comments have improved the paper in many ways. We hope you agree that the revised version builds a richer and stronger contribution.

We have rewritten the paper, improved the English, incorporated some paragraphs, and developed one figure, along with new supporting references. We also took the opportunity to place the text according to the new MDPI template.

For a better understanding of the new version of the manuscript, all responses have been highlighted in the text with a yellow color.

The following page deal with the detailed comments you raised.

Reviewer #1 – round 1

  1. The contribution of the paper is missing. Please clearly itemize the paper contribution in the introduction section.
  2. The figures' titles are ambiguous and need more specified explanations.
  3. For the proposed conceptual model to be more effective, some experimental experiences are required, such as information gathered from questionnaires completed by stakeholders.

Response:

Thank you for your insightful recommendations!

  1. For a better understanding of the new version of the manuscript, this response has been highlighted in the text with a yellow color. We added paragraphs in lines 129 to 134.
  2. This answer has been highlighted in the text with a yellow color in lines 173, 244-245, 522.
  3. Jabareen [15] mentions that the conceptual framework should make sense not only to the researcher but also to other scholars and practitioners. In this sense, the validation of the CM is a process that involves the researcher and then among “outsiders”. The CM was not validated yet, this process will be done on next paper. We justify that other stakeholder will validate the CM in lines 28-29 in Abstract section, pointing that: “The CM will be validated and applied in a manufacturing industry on next publication”.

Author Response

General comments

First, we would like to thank you for the time devoted in providing insightful recommendations. We believe that your comments have improved the paper in many ways. We hope you agree that the revised version builds a richer and stronger contribution.

We have rewritten the paper, improved the English, incorporated some paragraphs, and developed one figure. We also took the opportunity to place the text according to the new MDPI template.

For a better understanding of the new version of the manuscript, all responses have been highlighted in the text with a blue color.

The following page deal with the detailed comments you raised.

Reviewer #2 – round 1

  1. The result of the study has not been  stated in the abstract yet. After this sentence it supposed to be the explanation of the result.
  2. It could be better if the authors can explain that in the second part consist of four steps.
  3. And it much better if the author can present the methodology in a diagram so it will be easier for reader to understand it.

Response:

Thank you for your insightful recommendations!

  1. For a better understanding of the new version of the manuscript, this response has been highlighted in the text with a blue color. We added paragraphs in lines 25 to 28.
  2. We itemize with numbers to facilitate the understanding. This answer has been highlighted in the text with a blue color in lines 160 to 170.
  3. We created a Figure to represent the CM development methodology, which is presented below the line 172.

Reviewer 3 Report

I am happy to see the document. I can clearly understand the motivation of this paper, and a comprehensive and coherent presentation.

The authors have followed logical and scientifically-approved steps to conduct their research, and the studied publications have been evaluated in a correct and profound manner.

Most of the references are applicable and lately published (in 2019, or 2020). In addition, the authors have justified sufficiently why they rely upon some old references.

But, the manuscript is still very poorly written and filled with typos, although they claimed they had addressed some of the issues.

Author Response

General comments

First, we would like to thank you for the time devoted in providing insightful recommendations. We believe that your comments have improved the paper in many ways. We hope you agree that the revised version builds a richer and stronger contribution.

We have rewritten the paper, improved the English, incorporated some paragraphs, and developed one figure. We also took the opportunity to place the text according to the new MDPI template.

The following page deal with the detailed comments you raised.

Reviewer #3 – round 1

  1. The manuscript is still very poorly written and filled with typos, although they claimed they had addressed some of the issues.

Response:

Thank you for your insightful recommendations!

  1. We have improved the English of the manuscript.

Reviewer 4 Report

This is an interesting study for IoT technologies survey. This study can be perfect research if the definition of the PRISMA can be clarified in the introduction. In the survey result needs more detail description.

Author Response

General comments

First, we would like to thank you for the time devoted in providing insightful recommendations. We believe that your comments have improved the paper in many ways. We hope you agree that the revised version builds a richer and stronger contribution.

We have rewritten the paper, improved the English, incorporated some paragraphs, and developed one figure. We also took the opportunity to place the text according to the new MDPI template.

For a better understanding of the new version of the manuscript, all responses have been highlighted in the text with a green color.

The following page deal with the detailed comments you raised.

Reviewer #4 – round 1

This is an interesting study for IoT technologies survey. This study can be perfect research if the definition of the PRISMA can be clarified in the introduction. In the survey result needs more detail description.

  1. (…) definition of the PRISMA can be clarified in the introduction.
  2. In the survey result needs more detail description.

Response:

Thank you for your insightful recommendations!

  1. The PRISMA was clarified in the Introduction and in the CM development methodology. For a better understanding of the new version of the manuscript, this response has been highlighted in the text with a green color. We added paragraphs in lines 137-138, 156-157, 177 to 180.
  2. This answer has been highlighted in the text with a green color in lines 281-282 and 508 to 509.

Round 2

Reviewer 1 Report

The authors applied almost all the comments, however; representing itemized contributions, making boxes in the figures colorful and a slight expansion of explanation about future work in the conclusion, or even adding a section for future work will motivate more readers to consider the paper.

Author Response

General comments

Thank you for your insightful recommendations! The responses have been highlighted in the text with a yellow color. The following page deal with the detailed comments you raised.

Reviewer #1 – round 2

The authors applied almost all the comments, however; …

  1. … representing itemized contributions,
  2. … making boxes in the figures colorful and a
  3. … slight expansion of explanation about future work in the conclusion, or even adding a section for future work will motivate more readers to consider the paper.

Response:

  1. This response has been highlighted in the text with a yellow color in lines 636 to 651.
  2. This answer has been viewed in the Figures 1, 2 and 3.
  3. This answer has been highlighted in the text with a yellow color in lines 141, 655 to 689.

Reviewer 3 Report

Now paper can be accepted 

Author Response

Thank you for your insightful recommendations! 
